# Importance of feeding status evaluation in older patients undergoing hemodialysis

**Satoko Notomi[1], Mineaki Kitamura** [1,2] *, **Noriko Horita[3], Kosei Yamaguchi**[1,2], **Takashi Harada[1], Tomoya Nishino[2], Satoshi Funakoshi[1], Yasuyo Abe**[3]

**1** Nagasaki Renal Center, Nagasaki, Japan, **2** Department of Nephrology, Nagasaki University School of Medicine Graduate School of Biomedical Sciences, Nagasaki, Japan, **3** Nishikyushu University Graduate School of Human Care Sciences, Saga, Japan

* minekitamura@nagasaki-u.ac.jp

## Abstract

Older hospitalized patients undergoing hemodialysis are increasingly experiencing malnutrition caused by dysphagia. However, only a few studies have focused on this problem. We used the Kuchikara Taberu Balance Chart (KTBC) to evaluate the patients' feeding status and examined its association with their nutritional status and prognosis. This study included elderly patients undergoing hemodialysis who were hospitalized at Nagasaki Renal Center for > 3 days between June 2021 and February 2022. In total, 82 inpatients were included [mean age, 73.4 ± 10.0 years; men, 57.3%; median dialysis vintage, 79.0 months (interquartile range, 37.3–164.8)]. We classified patients with lower than the median KTBC score (57 points) as being at risk for dysphagia; 37 patients (45.1%) were at risk for dysphagia. Spearman's rank correlation coefficient ($\rho$) elucidated that the KTBC total score was significantly associated with each nutritional indicator [serum albumin level ($\rho$ = 0.505, p < 0.001); geriatric nutritional risk index ($\rho$ = 0.600, p < 0.001); and nutritional risk index ($\rho$ = -0.566, p < 0.001)]. The KTBC score was also closely associated with the body mass index ($\rho$ = 0.228, p = 0.04). Patients with a lower KTBC score showed poor prognosis (log-rank test: p = 0.001), and age- and sex-adjusted Cox proportional analysis showed that the KTBC score was associated with life prognosis (hazard ratio, 0.90; 95% confidential interval, 0.86–0.94; p < 0.001). Therefore, we concluded that the patients at risk of dysphagia, identified using the KTBC score, were malnourished and had a poor prognosis. Hence, the evaluation of dysphagia using the KTBC is encouraged to prevent malnutrition in vulnerable older patients undergoing hemodialysis.

**Data Availability Statement:** All relevant data are within the paper and its Supporting information files.

## Introduction

Nutritional problems have been highlighted in older patients undergoing hemodialysis in Japan [1]. Among diseases and health complications in older adults, stroke and dementia can evoke high rates of dysphagia [2], which is known to be associated with malnutrition [3]. These complications and mental problems (i.e., dementia) of older patients undergoing hemodialysis [4–7] can also have a negative impact on activities of daily living, feeding

**Funding:** The authors received no specific funding for this work.

**Competing interests:** The authors have declared that no competing interests exist.

status, and nutritional status, thus, resulting in poor prognosis [8, 9]. However, the dysphagia status has not been commonly evaluated at dialysis facilities compared to nursing homes. It would be important to conduct feeding status evaluation to prevent malnutrition at dialysis facilities.

The Kuchikara Taberu Balance Chart (KTBC) is a questionnaire-based evaluation tool for feeding status, including dysphagia, developed in Japan in 2015 [10]. The KTBC has been previously shown to be reliable and valid for the evaluation of swallowing [11]. It consists of 13 factors related to physical, nutritional, and medical conditions [12], and parameters that can be evaluated objectively based on medical records, such as fever, times of suction, use of artificial teeth, need for assistance during meals, and food modification, were used for evaluation [10]. Thus, the KTBC is non-invasive [11] and easy to use for medical staffs. However, there are no previous studies evaluating the feeding status of older patients undergoing hemodialysis with KTBC.

We hypothesized that dysphagia is associated with poor prognosis via malnutrition in older patients undergoing hemodialysis. This study aimed i) to examine the feeding status, including dysphagia, using the KTBC, and elucidate the factors related to dysphagia; and ii) to reveal the association between the feeding and nutritional status and life prognosis.

## Materials and methods

### Study design

This is a retrospective study from the time of KTBC evaluation to the end of February 2022. We examined the association between feeding status evaluated by KTBC, nutritional status, and life prognosis. This study was approved by the Institutional Review Board (22001) of Nagasaki Renal Center (Nagasaki, Japan) and was conducted following the 1964 Declaration of Helsinki and its subsequent amendments. Although the patients in this study were informed of their participation, the ethics committee waived the requirement for consent based on the retrospective nature of the analysis and the anonymization of the patients' data.

### Participants

This study included patients undergoing hemodialysis who were admitted to the Nagasaki Renal Center (Nagasaki, Japan) during the observation period. Patients with dialysis duration $\geq$ 3 months were included. Patients who were hospitalized for chemotherapy, those who left our facility before the evaluation of dysphagia, those hospitalized for $<$ 3 days, those on enteral nutrition, and those who were transferred to another hospital within 1 month were excluded. The KTBC was evaluated at least 3 days after admission to hospital. For patients with multiple hospitalizations during the observation period, the evaluation conducted at the first admission was used in this study.

### Data collection

Patient characteristics, including age, sex, duration of dialysis, and health complications, such as diabetes mellitus, ischemic heart disease, and arteriosclerosis obliterans, were obtained from description of pre-existing diseases in medical records. Dementia was determined by assessing Mini-Mental State Examination [13] or according to a previous diagnosis. Nutritional indicators, namely the serum albumin level, GNRI, and NRI were obtained from routine blood examinations conducted every month. To examine the association with the KTBC score and nutritional indicators briefly, we focused on the nutritional indicators of the closest month at the time of KTBC evaluation. Blood tests were routinely conducted at the beginning of dialysis

treatment (not fasting). Serum albumin was analyzed using the bromocresol green method. According to the protein energy wasting (PEW) criteria, reported by the International Society of Renal Nutrition and Metabolism, serum albumin levels < 3.8 g/dL indicate the risk of malnutrition [14]. The GNRI was calculated based on patient serum albumin levels and body weight using a modified formula proposed by Yamada, et al. (2008): GNRI = [14.89 × albumin (g/dL)] + [41.7 × (body weight/ideal body weight); for Body Mass Index (BMI) > 22 kg/m$^2$, body weight/ideal body weight = 1] [15]. A GNRI < 91.2 was defined as risk of malnutrition [15]. The NRI was evaluated based on age, serum albumin level, BMI, total cholesterol, and serum creatinine levels, as described in a previous study [16]. An NRI $\geq$ 11 was defined as high risk of malnutrition [16]. The KTBC was used to evaluate the dysphagia of inpatients, with each factor rated from 1 (worst) to 5 (best) points [11]. The evaluation criteria for each factor were defined by the developer of the KTBC [10]. Dysphagia was evaluated by a trained registered dietitian who was blinded to blood examination data. Patients were divided into two groups based on their median KTBC total score. In addition to the KTBC total score, we focused on six major factors in the KTBC score ("overall condition," "cognitive function while eating," "severity of pharyngeal dysphagia," "position and endurance while eating," "food modification," and "nutrition").

## Statistical analysis

Categorical values are shown as numbers (in %), and continuous variables are shown as mean ± standard deviation or median values with interquartile ranges. The Wilcoxon Sum Rank test was used to compare the two groups. Spearman's rank correlation coefficient ($\rho$) was used to examine the association between the KTBC score and nutritional parameters such as serum albumin level, GNRI, NRI, and monthly BMI decrease. A logistic regression analysis was conducted to elucidate the association between a lower KTBC total score (less than median) and patient background factors, such as age, sex, dialysis vintage, and health complications. A multivariable logistic regression analysis was conducted using the parameters with $p < 0.05$ in the univariable logistic regression analysis. Kaplan–Meier curves were used to show the survival rate for the two groups, and the log-rank test was used to compare the groups. Cox proportional analysis was used to evaluate the risk of mortality, including the KTBC total score, age, sex, dialysis vintage, and health complications. Age- and sex-adjusted Cox proportional hazards analyses were also performed. Statistical significance was set at $p < 0.05$. All statistical analyses were performed using JMP 15 software (SAS Institute Inc., Cary, NC, USA).

## Results

### Patient background

During the observation period, 150 patients undergoing hemodialysis were admitted to Nagasaki Renal Center. After excluding 68 patients, 82 inpatients (age 73.4 ± 10.0 years; male, 57.3%) were analyzed. The patient information flow chart is presented in Fig 1. Major reasons for hospitalization were difficulty in moving and staying at home (n = 36), continued pneumonia (n = 6), fever (n = 5), and enteritis (n = 5). The median KTBC total score was 57 (interquartile range: 50–61) points; 37 patients had a lower score than the median KTBC total score. The demographic features of the patients divided by a KTBC score of 57 points are shown in Table 1. There were significant differences in stroke history, dementia, BMI, serum albumin level, GNRI, and NRI between the two groups.

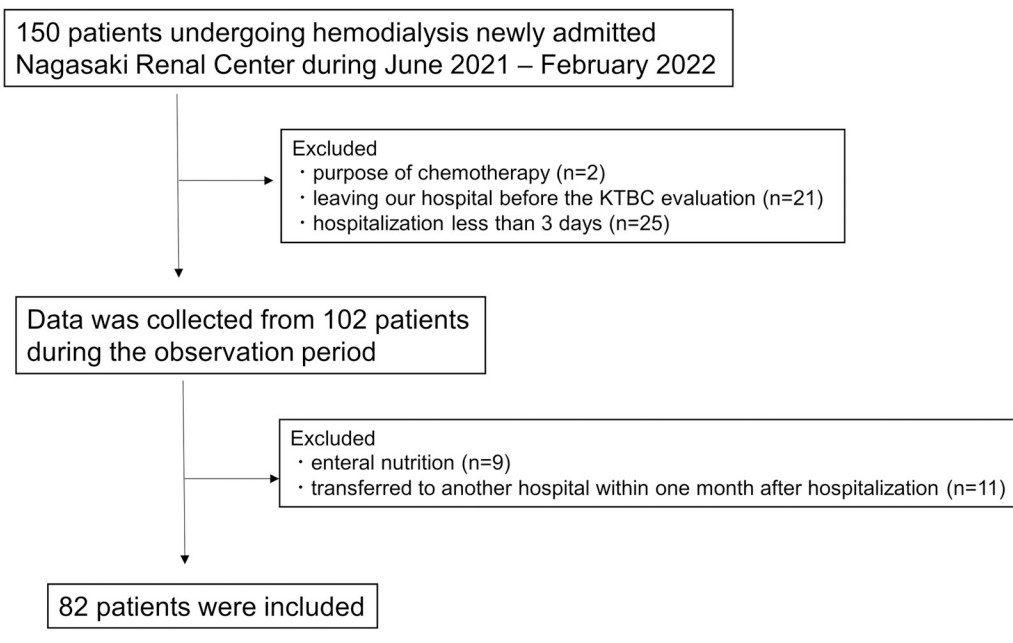

**Fig 1. Patients' flow chart.**

### Feeding status evaluated using the KTBC and its association with nutritional status or life prognosis

The histogram of the KTBC total score is shown in Fig 2. Approximately one-third of our participants had a total score > 60 points.

The KTBC total score was significantly correlated with serum albumin level ($\rho = 0.505$, $p < 0.001$), GNRI ($\rho = 0.600$, $p < 0.001$), and NRI ($\rho = -0.566$, $p < 0.001$; Fig 3a–3c). Of the 82

**Table 1. Summary of patient background.**

| | Total (n = 82) | KTBC total score ($<$ 57) (n = 37) | KTBC total score ($\geq$ 57) (n = 45) | p |
|---|---|---|---|---|
| Age (years) | 73.4 ± 10.0 | 74.5 ± 12.3 | 72.5 ± 7.8 | 0.44 |
| Sex, n (%) | | | | |
| Male | 47 (57.3%) | 22 (59.5%) | 25 (55.6%) | 0.72 |
| Female | 35 (42.7%) | 15 (40.5%) | 20 (44.4%) | |
| Dialysis vintage (months) | 79.0 (37.3–164.8) | 57.0 (38.0–120.0) | 95.0 (34.5–193.5) | 0.13 |
| Diabetes mellitus history | 38 (46.3%) | 20 (54.1%) | 18 (40.0%) | 0.20 |
| Ischemic heart disease history | 34 (41.5%) | 15 (40.5%) | 19 (42.2%) | 0.88 |
| Stroke history | 18 (22.0%) | 12 (32.4%) | 6 (13.3%) | 0.04 |
| Arteriosclerosis obliterans | 17 (20.7%) | 7 (18.9%) | 10 (22.2%) | 0.71 |
| Dementia | 22 (26.8%) | 15 (40.5%) | 7 (15.6%) | 0.01 |
| Serum albumin level (g/dL) | 3.1 (2.7–3.3) | 2.8 (2.6–3.2) | 3.2 (3.0–3.6) | < 0.001 |
| Geriatric nutritional risk index | 82.4 ± 9.35 | 77.45 ± 9.73 | 86.50 ± 6.74 | < 0.001 |
| Nutritional risk index | 9.0 (5.8–11.0) | 11.0 (9.0–12.0) | 8.0 (4.5–9.0) | < 0.001 |
| Body mass index | 20.2 (17.8–21.8) | 18.3 (16.5–20.8) | 21.3 (19.1–22.0) | 0.001 |
| Kuchikara Taberu Balance Chart total score | 57.0 (50.0–61.0) | 49.0 (41.5–53.0) | 60.0 (58.0–62.0) | < 0.001 |

Wilcoxon sum test, paired-t-test and chi-square test were used for analysis.

Number of patients

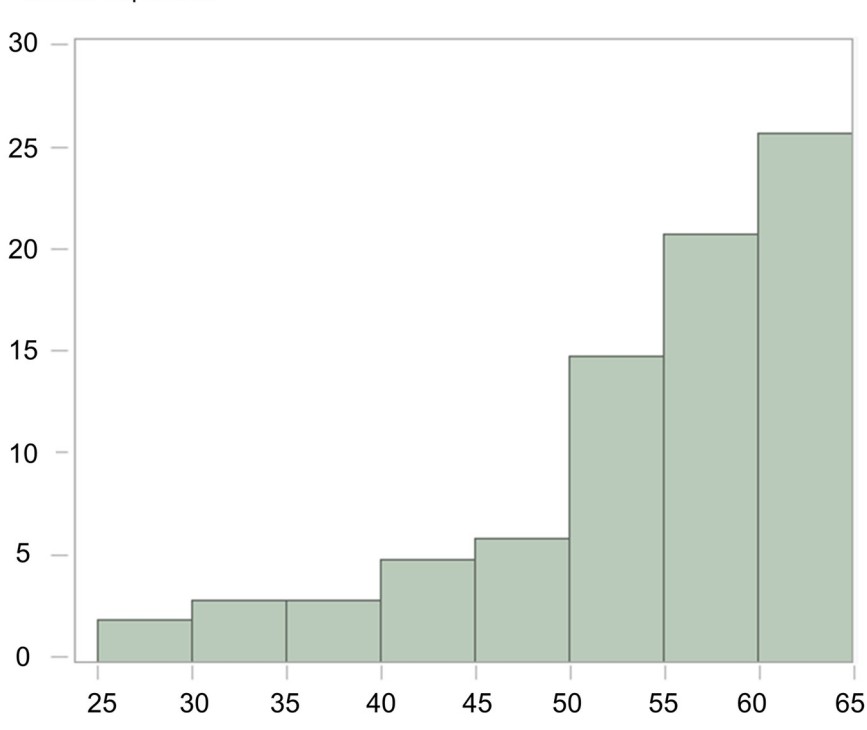

**Fig 2. Histogram of the KTBC total score.**

patients, 76 (92.7%) were suspected of having malnutrition according to their serum albumin levels (Fig 3a), 67 (81.7%) were defined as being at risk of malnutrition per the GNRI (Fig 3b), and 32 (39.0%) were defined as being at risk of malnutrition per the NRI (Fig 3c). Additionally, the KTBC total score was significantly associated with monthly decrease in BMI ($\rho$ = 0.228, p = 0.04; Fig 4).

Univariable logistic regression analysis for a lower KTBC score was conducted based on patient characteristics. BMI, stroke history, and dementia were significantly associated with KTBC (p < 0.05). Next, we conducted a multivariable logistic regression analysis including these parameters; lower KTBC scores were associated with BMI only (odds ratio, 0.79; 95% confident interval 0.66–0.93, p = 0.004; Table 2).

The median observation period was 118 days (interquartile range, 71–217 days). Of 82 patients, 25 died during the observation period. In our participants, cardiovascular disease was the most common cause of death (32.0%), followed by infectious complication (12.0%) and ileus (8.0%). Other major causes of death include cancer, multiple organ failure, and aspiration pneumonia, each accounting for 4.0% of cases of death. A log-rank analysis showed that patients with lower KTBC scores had poorer prognosis than those with higher scores (p = 0.001; Fig 5). Univariable Cox proportional analysis indicated that age, BMI, dementia, and KTBC score were associated with patient prognosis. Further, age- and sex-adjusted Cox proportional analysis showed that the BMI (hazard ratio, 0.86; 95% confidence interval, 0.75–0.99; p = 0.04) and the KTBC total score (hazard ratio, 0.90; 95% confidence interval, 0.86–0.94; p < 0.001) were associated with life prognosis (Table 3).

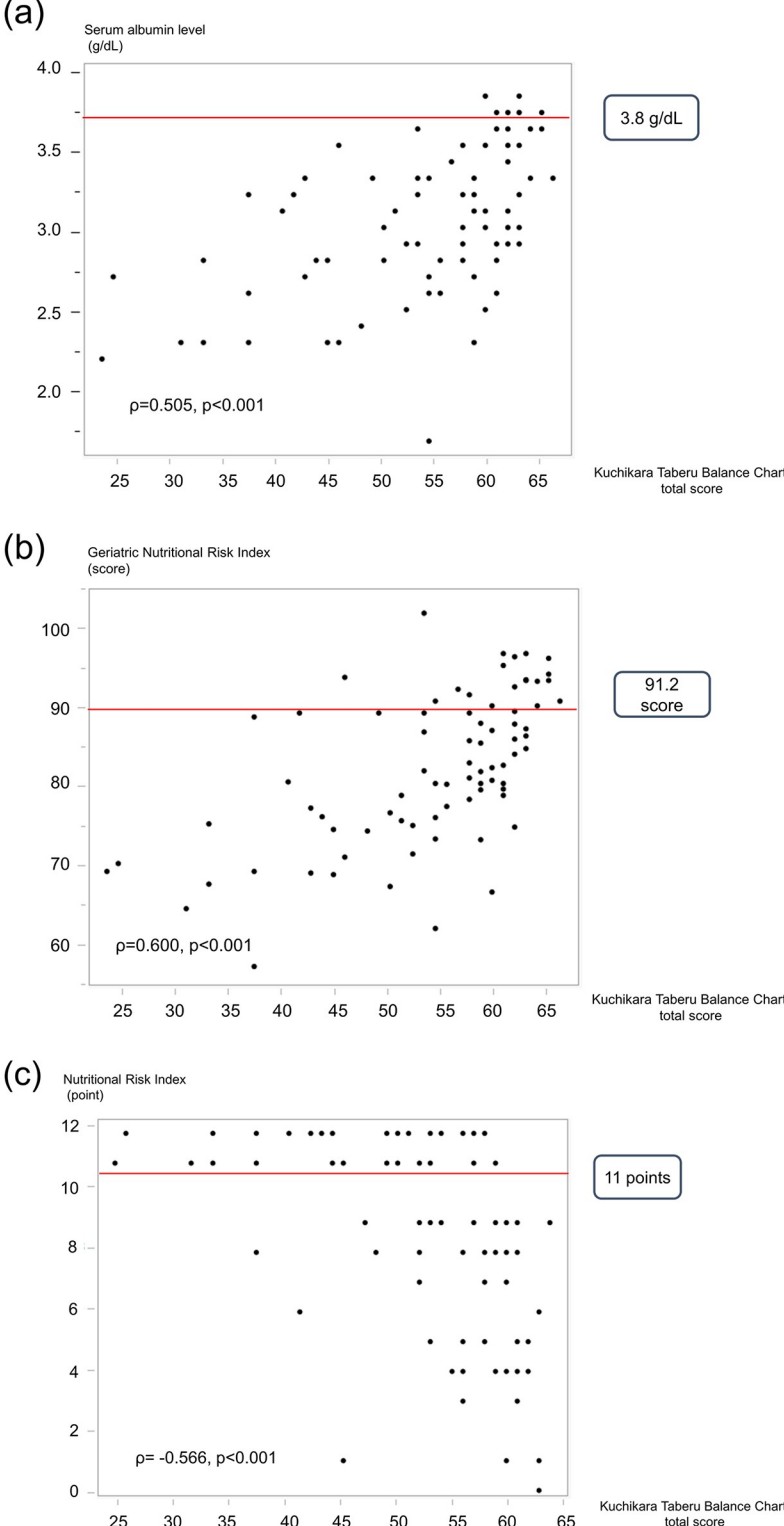

**Fig 3. Association between the KTBC total score and nutritional status; a. serum albumin level, b. GNRI, and c. NRI.**

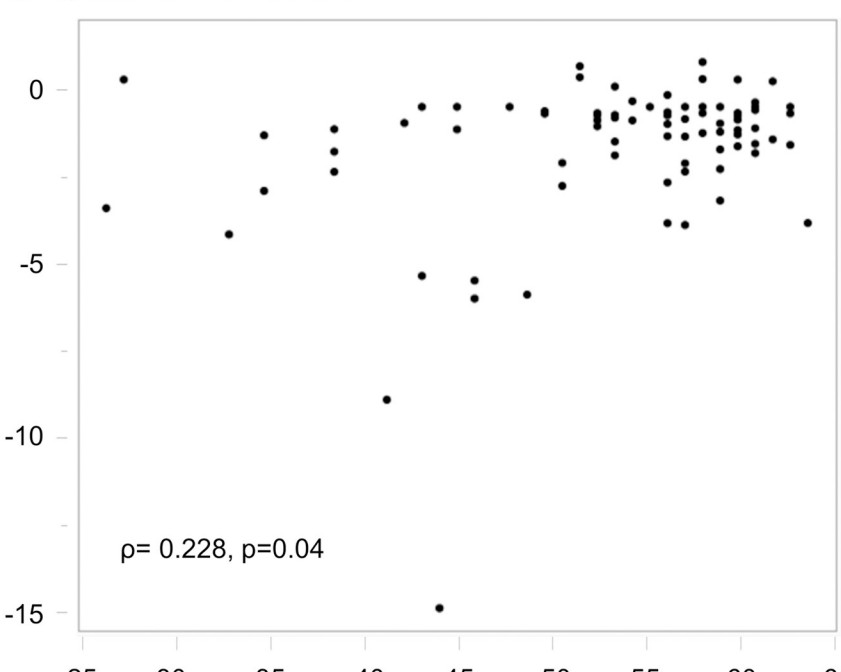

**Fig 4. Association between the KTBC total score and percentage BMI decrease.**

## Association between major factors of the KTBC and dementia or nutritional indicators

Dementia was significantly associated with feeding problems, such as cognition (p = 0.01), swallowing (p < 0.001), and posture maintenance during meals (p = 0.01; S1 Table). Among the major factors of the KTBC, food modification was strongly associated with all nutritional indicators (p < 0.001 for serum albumin level, GNRI, and NRI). Serum albumin level was more highly associated with over-all condition (p = 0.001) than with nutrition (p = 0.07; S2 Table).

**Table 2. Logistic regression models for lower Kuchikara Taberu Balance Chart (< 57 points).**

|  | Univariable | | | | Multivariable | | | |
|---|---|---|---|---|---|---|---|---|
|  | OR | 95%CI | | p | OR | 95%CI | | p |
|  |  | Lower | Upper |  |  | Lower | Upper |  |
| Age per 1 year old | 1.02 | 0.98 | 1.07 | 0.37 |  |  |  |  |
| Male vs. Female | 1.17 | 0.49 | 2.86 | 0.72 |  |  |  |  |
| Dialysis vintage per 1 month | 0.96 | 0.92 | 1.01 | 0.10 |  |  |  |  |
| Body Mass Index per 1 kg/m$^2$ | 0.77 | 0.64 | 0.90 | 0.001 | 0.79 | 0.66 | 0.93 | 0.004 |
| Diabetes mellitus history | 1.76 | 0.74 | 4.31 | 0.20 |  |  |  |  |
| Ischemic heart disease history | 0.93 | 0.38 | 2.26 | 0.88 |  |  |  |  |
| Stroke history | 3.12 | 1.07 | 9.97 | 0.04 | 2.69 | 0.83 | 9.41 | 0.10 |
| Arteriosclerosis obliterans | 0.82 | 0.27 | 2.39 | 0.71 |  |  |  |  |
| Dementia | 3.70 | 1.35 | 11.02 | 0.011 | 2.86 | 0.95 | 9.17 | 0.06 |

OR, odds ratio; CI, confidence interval.

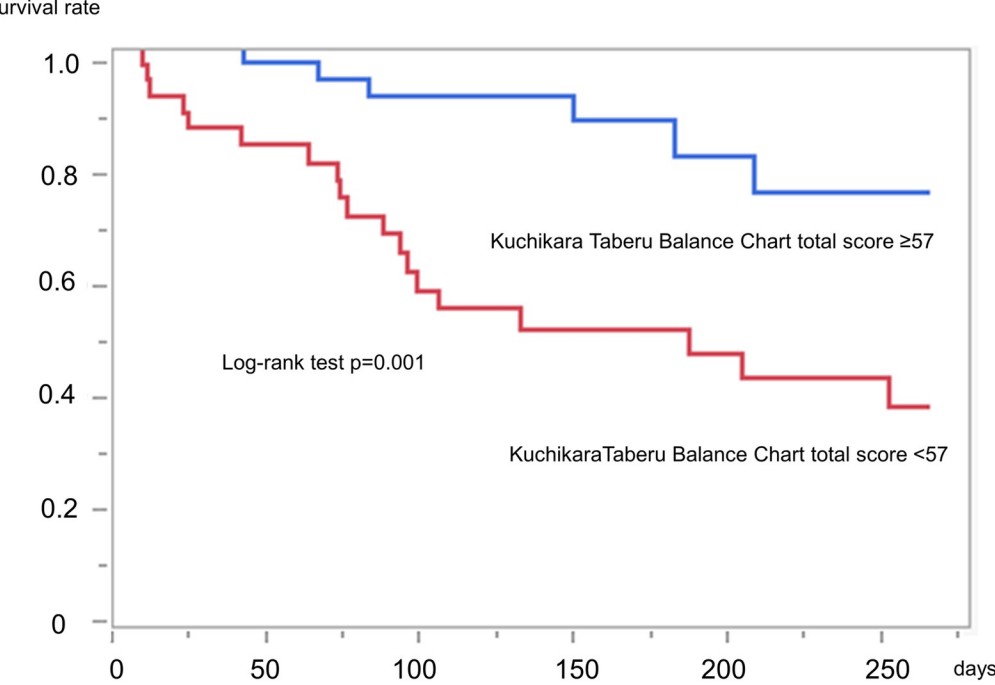

**Fig 5. Kaplan–Meier analysis of participants.**

## Discussion

To our knowledge, this is the first study that evaluated the feeding status, including dysphagia, of older patients undergoing hemodialysis using the KTBC. The KTBC has several advantages as it is non-invasive, easy to evaluate, and has an accuracy equivalent to videoendoscopic examination of swallowing function. High sensitivity of KTBC with respiratory conditions and postural endurance have been reported in a previous study [17]. As our older patients had many difficulties related to maintenance of posture due to several health complications, the KTBC was considered a suitable tool to verify the effect on posture maintenance on feeding

**Table 3. Cox regression analysis for all-cause mortality.**

| | Univariable | | | | Age sex ajusted | | | |
|---|---|---|---|---|---|---|---|---|
| | HR | 95%CI | | p | HR | 95%CI | | p |
| | | Lower | Upper | | | Lower | Upper | |
| Age per 1 year old | 1.05 | 1.01 | 1.10 | 0.01 | 1.05 | 1.01 | 1.10 | 0.01 |
| Male v.s. Female | 1.41 | 0.62 | 3.20 | 0.41 | 1.42 | 0.63 | 3.22 | 0.40 |
| Dialysis vintage per 1 month | 1.01 | 0.97 | 1.04 | 0.69 | 1.04 | 1.00 | 1.09 | 0.07 |
| Body Mass Index per 1 kg/m² | 0.85 | 0.75 | 0.97 | 0.01 | 0.86 | 0.75 | 0.99 | 0.04 |
| Diabetes mellitus history | 0.78 | 0.35 | 1.72 | 0.54 | 0.95 | 0.41 | 2.16 | 0.90 |
| Ischemic heart disease history | 0.88 | 0.40 | 1.97 | 0.73 | 0.91 | 0.41 | 2.04 | 0.82 |
| Stroke history | 0.69 | 0.26 | 1.85 | 0.46 | 0.83 | 0.30 | 2.26 | 0.71 |
| Arteriosclerosis obliterans | 0.88 | 0.35 | 2.21 | 0.78 | 1.06 | 0.39 | 2.89 | 0.91 |
| Dementia | 2.46 | 1.12 | 5.40 | 0.03 | 2.03 | 0.83 | 4.97 | 0.12 |
| Kuchikara Taberu Balance Chart total score | 0.90 | 0.86 | 0.93 | < 0.001 | 0.90 | 0.86 | 0.94 | < 0.001 |

HR, hazard ratio; CI, confidence interval.

status. Although we focused on elderly patients in this study, 11 patients (13%) were under 60 years old because all patients hospitalized during the observation period were included. As the mean age of our participants was over 70 years old, we assume that the result of this study reflected the actual feeding status of elderly patients undergoing hemodialysis.

Approximately one-third of our participants had a score > 60 points (90% of the KTBC total scores), indicating that the feeding status of our participants was generally good. However, this study revealed that patients with lower KTBC scores, who were at risk of dysphagia, were undernourished, and had a poor prognosis; these findings were consistent with the findings of previous studies [18, 19].

Our study elucidated the significant association between the KTBC scores and BMI. This association can be explained by two possible mechanisms. One aspect is low BMI as a cause of poor feeding status; BMI is related with swallowing function that is associated with muscle mass. Previous studies have suggested that patients undergoing hemodialysis tend to lose their muscle mass, which leads to weakened swallowing function [20–22]. A decrease in muscle mass results in weakness [23] and loss of mobility [24]; these changes can negatively correlate with the effective flow of swallowed materials [2]. Therefore, a lower BMI can lead to poor feeding status. Another aspect is low BMI as a result of poor feeding status; patients with feeding difficulties consume limited food and liquid [2] leading to energy and protein shortages that further lead to malnutrition and a lower BMI. Considering these two aspects, patients with a lower BMI and risk of dysphagia consume less food, resulting in further weight loss. Thus, dysphagia is a gateway to a vicious circle of malnutrition leading to poor prognosis. Our participants were older; approximately one third of the patients were aged > 80 years. The elderly population is easy to be in "frail" situation, which is known as a state of vulnerability to poor homeostasis resolution after a stressor event [25]. Hospitalization may be a stressor to our older participants, and the patients were more likely to be in a vicious cycle of malnutrition related to dysphagia.

For patients undergoing hemodialysis, a lower BMI is known to be a greater mortality risk factor [26], and unlike in the general population, a higher BMI is associated with survival advantage; this is known as "reverse epidemiology" [27]. Although the cause of this reverse epidemiology has not been fully elucidated, one of the reasons could be that patients with a lower BMI are more susceptible to inflammation leading to a type of malnutrition, called PEW [26]. PEW evokes disordered catabolism, thus, leading to muscle loss [28]. Malnutrition and muscle loss affect cardiovascular disease negatively, which is a major cause of mortality in patients undergoing hemodialysis [29]. Additionally, as the immune system is susceptible to malnutrition, PEW can affect infectious complications negatively, which is also a major cause of mortality [30]. Our older "frail" hospitalized patients with low BMI were easy to be in PEW situation, and muscle loss leading to dysphagia was likely to occur. Moreover, dysphagia can cause further weight loss. In our participants, cardiovascular disease was the most common cause of death, followed by infectious diseases. Aspiration pneumonia was also one of the causes of death. This may suggest that a low BMI related to dysphasia was associated with death and life prognosis. In our participants, the median BMI was 20.2 kg/m$^2$; in few individuals with obesity, however, a higher BMI with better feeding status was associated with higher survival rates. Although previous studies on reverse epidemiology have not discussed the nutritional status of the elderly, dysphagia in older patients undergoing hemodialysis might be considered one of the appearances of reverse epidemiology. Therefore, the evaluation of dysphagia is crucial to maintain good life prognosis.

Patients with dementia tended to have a poor feeding status related to poor cognition, swallowing, and posture maintenance conditions. Loss of appetite, mastication disorder, and dysphagia caused by dementia delay oral transit time [31] and evoke difficulties in self-feeding

[2]. Our participants were old, 26.8% of them had dementia, and many of them were classified as having poor feeding status. According to the description of medical records, most of the patients with dementia need assistance to eat. As stated in literature, difficulties in oral intake and self-feeding may occurred in patients with dementia, and these could be difficulty in eating independently. With the aging process in patients undergoing hemodialysis, effective measures against dysphagia caused by dementia should be implemented at dialysis facilities.

In addition to BMI, nutritional indicators, such as serum albumin levels, GNRI, and NR, were significantly correlated with the KTBC total score. Examining the association between KTBC major factors and nutritional indicators, the characteristics of each indicator are explained as follows. A previous study suggested that the serum albumin level reflected malnutrition and protein loss caused by catabolism and inflammation [32]. In this study, the serum albumin levels were more strongly associated with over-all condition than with nutrition. Hence, it could be possible to use the serum albumin level as not only a nutritional indicator but also an indicator of pathophysiological status. In this study, the GNRI and the NRI reflected the feeding status almost equally.

This study revealed that patients at risk of dysphagia were undernourished and had a poor prognosis. Based on the evaluation of KTBC, the following measures may be important. First, the management of posture during meals for patients with dementia. Our patients with dementia had difficulty maintaining the posture, leading to a poor feeding status. Previous studies have suggested that chin tuck or head tilt posture maintenance changes the speed and flow direction of food and improves the safety of swallowing [33, 34]. This could be one of the approaches to prevent aspiration. Second, the improvement of paste meals for patients at risk of dysphagia. This study revealed that "food modification" was strongly associated with malnutrition. In KTBC, patients who eat paste meals are rated one (worst); that means patients who eat paste meals tend to have malnutrition. Paste meals contain a relatively large amount of water to enable stirring and swallowing; consequently, substantial amounts of ingredients decreased, leading to reduced energy and protein intake. A previous study suggested that adding oil to food could be an alternative nutritional method for patients with dysphagia [35], as lipids are high in calories, and suitable for adjusting the rheological properties of foods [35]. Further studies are needed to verify the effect of adding oil to food. Third, the examination of appropriate nutritional supplements for patients at risk of dysphagia. This study revealed that maintaining optimal BMI is an advantage for survival, however our participants without enteral nutrition showed a -0.48% (-1.44–0%) decrease in BMI during the observation period. Although we excluded patients on enteral nutrition from the analysis, they tended to have a lower rate of BMI decrease despite the lower KTBC total score [BMI decrease, -0.09% (-0.79–0.41%); KTBC total score, 27 (22–40) points]. This means that enteral nutrition is an alternative method that can supply adequate energy and protein for patients at risk of dysphagia. Fourth, the improvement of the swallowing function. Several studies have reported that swallowing function was improved by interventions, such as oral care, for issues revealed by KTBC evaluation [12, 36–38]. Recent evidence suggests that swallowing rehabilitation and early preventive efforts are essential for preventing dysphagia. Intervention studies have examined improvements in life prognosis by nutritional intervention and swallowing rehabilitation [39, 40]. As this is a retrospective study based on clinical practice, it was difficult to standardize the timing of KTBC evaluation. It would be desirable to establish a protocol for early evaluation and intervention during hospitalization to prevent malnutrition.

However, this study has several limitations. As our participants were limited to older patients undergoing hemodialysis, the results may have been influenced by the participants' characteristics. Additionally, the association between the feeding and nutritional status was examined in a cross-sectional manner. To apply the results of this study to the general elderly

population, evaluation of dysphagia using the KTBC in a larger sample at diverse facilities is needed. We used the BMI as a nutritional indicator related to dysphagia; however, we did not consider body fluid removal during the dialysis session. Therefore, the results did not reflect the body composition or edema status of patients. Additionally, as dysphagia is related to muscle capability, indicators reflecting muscle function should also have been considered.

## Conclusions

Patients at risk of dysphagia, as measured by the KTBC, have a poor prognosis. As the KTBC is associated with nutritional indicators, malnutrition has a negative impact on patient prognosis. As older patients undergoing hemodialysis are more vulnerable to dysphagia, the evaluation of dysphagia using the KTBC could be one of clinical assessment tools to prevent malnutrition.

## Supporting information

**S1 Data. Patients' data are supplied in a supporting information file.**
(XLSX)

**S1 Table. Association between major items of the Kuchikara Taberu Balance Chart and dementia.** Wilcoxon sum rank test was used.
(DOCX)

**S2 Table. Association between major items of the Kuchikara Taberu Balance Chart and nutritional indicators.** Spearman's rank correlation coefficient was used.
(DOCX)

## Acknowledgments

We convey our special thanks to healthcare workers for supporting this study.

## Author Contributions

**Conceptualization:** Satoko Notomi, Mineaki Kitamura, Yasuyo Abe.

**Data curation:** Satoko Notomi, Mineaki Kitamura, Kosei Yamaguchi.

**Formal analysis:** Satoko Notomi, Mineaki Kitamura, Kosei Yamaguchi, Yasuyo Abe.

**Investigation:** Satoko Notomi, Mineaki Kitamura.

**Methodology:** Satoko Notomi, Noriko Horita, Yasuyo Abe.

**Project administration:** Satoshi Funakoshi.

**Software:** Kosei Yamaguchi.

**Supervision:** Noriko Horita, Takashi Harada, Tomoya Nishino, Satoshi Funakoshi, Yasuyo Abe.

**Visualization:** Satoko Notomi, Yasuyo Abe.

**Writing – original draft:** Satoko Notomi, Mineaki Kitamura, Yasuyo Abe.

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
