## [Decision Letter · Decision Letter 0]

6 Sep 2022

PONE-D-22-12985Impact of feeding status evaluated by the Kuchikara Taberu Balance Chart on nutrition and life prognosis in older patients undergoing hemodialysisPLOS ONE

Dear Dr. Kitamura,

Thank you for submitting your manuscript to PLOS ONE. After careful consideration, we feel that it has merit but does not fully meet PLOS ONE’s publication criteria as it currently stands. Therefore, we invite you to submit a revised version of the manuscript that addresses the points raised during the review process.

ACADEMIC EDITOR:After observing the reviewers' comments, and carrying out a new reading, I believe that the manuscript needs several urgent changes. In this way, we invite the authors to review the title, reinforce the idea of justification. Still, that the need to review the results and re-write several parts of the discussion in a more assertive way. In this way, we hope that you will take note of the reviewers' comments and try to respond to them promptly.

We look forward to receiving your revised manuscript.

Kind regards,

Leonardo Costa Pereira, Doctor

Academic Editor

PLOS ONE

Reviewers' comments:

Reviewer's Responses to Questions

**Comments to the Author**

1. Is the manuscript technically sound, and do the data support the conclusions?

Reviewer #1: Partly

Reviewer #2: Yes

2. Has the statistical analysis been performed appropriately and rigorously? 

Reviewer #1: Yes

Reviewer #2: Yes

3. Have the authors made all data underlying the findings in their manuscript fully available?

Reviewer #1: Yes

Reviewer #2: Yes

4. Is the manuscript presented in an intelligible fashion and written in standard English?

Reviewer #1: Yes

Reviewer #2: Yes

5. Review Comments to the Author

Reviewer #1: This manuscript describes the evaluation of patients’ feeding status, undergoing hemodialysis, from Kuchikara Taberu Balance Chart (KTBC) and its association with their nutritional status and prognosis. The authors verified interesting associations and results of utilization of KTBC.

In general, the manuscript is interesting and brings important information’s about the theme studied.

I have some suggestions about the manuscript.

Title

The title is long, I recommend shortening it.

Keywords

The keywords must be modified, the authors repeat some words from the title.

Introduction

In general, the introduction is good, but, according to me, could be improved, especially the first two paragraphs.

Material and Methods

It is important to note that in this section, all information must be very detailed.

Then this section should be revised and improved.

For example:

1) “Data Collection”: some important information is missing, such as: a) All the health complications assessed should be listed; b) how the “dementia” was assessed? From medical records? This information is unclear.

2) How long time/months have the nutritional indicators (serum albumin level, GNRI, and NRI were obtained from routine blood examinations conducted every month) been obtained? This information is in “Results” section, but it is part of the MM and should be further explained in this section.

Discussion

In general, this section should be improved.

The authors explain their results with the literature, but, in my opinion, the discussions should be extended with examples and comparisons in the literature, and the age condition should be further discussed.

The authors affirm that a limitation of this study is “Evaluation of dysphagia using the KTBC in a larger sample at multiple centers is needed to examine generalizability of the results. The timing of the evaluation varied depending on the case. It would have been ideal to perform a KTBC assessment of all patients within a certain period post admission in our hospital.” About this statement, I have some suggestions/questions:

- The authors could discuss this important point of view in the “discussion” section;

- Considering this limitation, is it prudent to conclude that “the evaluation of dysphagia using the KTBC is highly recommended to prevent malnutrition.”? I don’t think so.

Reviewer #2: I would like to congratulate the authors for their interesting work. This is a retrospective study and therefore limited to its design. Despite this, it is relevant and has been well conducted. This study will contribute to clinical practice and the attention of health professionals regarding nutritional status and dysphagia. However, I have several comments and believe that some issues could be clarified and some sections could be better organized.

TITLE

I think it would be interesting to clarify what the use of the word nutrition wants to convey. Nutritional status? Therefore, I suggest adjusting the title.

ABSTRACT

Adequate and concise. I suggest just mentioning in line 6 that they are elderly patients, and the percentage value in parentheses is equivalent to n=37 (line 10).

INTRODUCTION

I suggest deleting the part of the third paragraph about the EAT-10 related information. It can be included in the discussion section if the authors wish. I think it would be the most appropriate session, precisely because it was not used in the study and contributed to making the introduction unnecessarily long.

It is not usual to mention figures in the introduction section. The manuscript already has three tables and six figures. There is no need to exemplify the KTBC tool. If the authors wish to keep Figure 1, I suggest using the methods when citing the tool or presenting the study's radar chart in the results.

The fourth paragraph, it's confusing and repetitive. I recommend making the objectives clear to the reader, including that they are elderly patients.

MATERIALS AND METHODS

In the Study design section, some information presented is duplicated, such as the data collection period until February 2022 (page 5, line 18; page 6, line 6; page 9, line 6), as well as for the nutritional indicators on pages 6 and 7. These can be presented in the Data collection section. The Study design section is large and with varied information, not only about the study design. I recommend including a Participants section to present the sample and eligibility criteria.

I did not understand the inclusion criterion of being over 20 years old. Is this not a sample of elderly patients?

Authors should value the methods section and present missing information. In Data Collection, detail the sociodemographic and clinical characteristics collected. Nutritional indicators may or may not be part of a separate section, but blood tests conditions (e.g., fasting or not?) and analysis methods (e.g., albumin, colorimetric bromocresol green method?) must be described. Was there any care in measuring weight due to changes in blood volume and edema in this population?

Figure 2: There is no need to repeat the collection period in the third text box, as it was already presented in the first. If the KTBC was performed after three days of hospitalization, shouldn't the sample with hospitalization less than three days (n=25) from the second exclusion text box be in the first exclusion step? Before having your data collected?

Is it not possible to adopt a plausible cut-off point for the value of albumin in this population, as was done for the GNRI and the NRI?

In Statistical analysis, line 9 has a missing parenthesis.

RESULTS

In the Patient background section, Figure 2 is presented the methods and results. Some researchers and reviewers consider it as part of the method and others as a first result. I recommend that the authors choose which of the two sessions will mention in figure 2. In the results, there is no need to describe the figure step. It could be something more general.

The first mention in the text about Table 1 does not present the main information in the text (page 9, line 14). This content is out of place on page 11 (between lines 8 and 11). I recommend making a single mention of Table 1 with the main findings presented in a general way at the beginning of the results session.

Present in the text comments on Figure 3.

ALL FIGURES AND TABLES: Whenever possible, use a full title for clarity for the reader.

DISCUSSION

I recommend avoiding mentioning the tables and figures in this section.

On page 17 (line 11), include the BMI unit and avoid using pejorative or stigmatizing language (e.g., “few were obese” by a few individuals with obesity).

Regarding reverse epidemiology, I think it is necessary to consider the other side of studies related to the topic. Many are inconsistent, some use BMI as a marker of adiposity, which does not measure central adiposity. At the same time, we know that the major cause of mortality in this population is attributed to cardiovascular disease.

Regarding limitations, it is necessary to mention the limitation of the use of BMI and possibly the lack of an adequate parameter for more effective conclusions through the assessment of body composition and muscle function, whether strength or functional capacity of elderly individuals and its relationship with dysphagia.

6. PLOS authors have the option to publish the peer review history of their article (what does this mean?). If published, this will include your full peer review and any attached files.

Reviewer #1: No

Reviewer #2: No

---

## [Author Response · Author response to Decision Letter 0]

19 Sep 2022

AUTHORS’ RESPONSES TO REVIEWERS’ COMMENTS

Reviewer 1

We would like to thank Reviewer 1 for the time and effort taken to review our manuscript and for providing valuable comments and suggestions, which have considerably helped us improve our manuscript. We have made every effort to address the issues raised and to respond to all comments. The revisions are indicated in red font in the revised manuscript. Please, fine next a detailed, point-by-point response to the reviewer’s comments. We hope that our revisions will meet the reviewer’s expectations.

Title

The title is long, I recommend shortening it.

Response: Please note that we have shortened the title to convey our idea more clearly.

Keywords

The keywords must be modified, the authors repeat some words from the title.

Response: Following the reviewer’s suggestion, we have changed the title and reviewed the key words. We have tried to avoid using the same terms in the title. 

Introduction

In general, the introduction is good, but, according to me, could be improved, especially the first two paragraphs.

Response: We agree with the reviewer that the Introduction section of the previous version of our manuscript was redundant and too long. Therefore, we have revised it accordingly. In the first paragraph, we have summarized the complications of patients undergoing hemodialysis and excluded duplicate text. In the second paragraph, we have summarized the contents combined with the previous third paragraph and presented an overview of the KTBC. Moreover, we have moved the details of the KTBC to the Discussion section.

Material and Methods

It is important to note that in this section, all information must be very detailed.

Then this section should be revised and improved.

For example:

1) “Data Collection”: some important information is missing, such as: a) All the health complications assessed should be listed; b) how the “dementia” was assessed? From medical records? This information is unclear.

Response: Please note that we have collected information concerning diabetes mellitus history, ischemic heart history, stroke history, and arteriosclerosis from the description of pre-existing disease in medical records. 

Dementia was determined by assessing the Mini-Mental State Examination, which is known as a validated assessment tool, or according to a previous diagnosis. We have provided this information in the Materials and Methods section as follows:

 “Dementia was determined by assessing Mini-Mental State Examination [13] or according to a previous diagnosis.” (Page 6, lines 4–5)

2) How long time/months have the nutritional indicators (serum albumin level, GNRI, and NRI were obtained from routine blood examinations conducted every month) been obtained? This information is in “Results” section, but it is part of the MM and should be further explained in this section.

Response: The serum albumin levels were assessed monthly using routine blood tests. The GNRI and NRI were calculated using the results of monthly blood tests. To examine the association with the KTBC score and the nutritional status briefly, we focused on the nutritional indicators of the closest month at the time of KTBC evaluation. We have provided this information in the Materials and Methods section as follows:

 “To examine the association with the KTBC score and nutritional indicators briefly, we focused on the nutritional indicators of the closest month at the time of KTBC evaluation. Blood tests were routinely conducted at the beginning of dialysis treatment (not fasting). Serum albumin was analyzed using the bromocresol green method. According to the protein energy wasting (PEW) criteria, reported by the International Society of Renal Nutrition and Metabolism, serum albumin levels < 3.8 g/dL indicate the risk of malnutrition [14].” (Page 6, lines 7–13)

Discussion

In general, this section should be improved.

The authors explain their results with the literature, but, in my opinion, the discussions should be extended with examples and comparisons in the literature, and the age condition should be further discussed.

Response: We agree with the reviewer that we should have extended the Discussion section. Following the reviewer’s suggestion, we compared the findings from the literature with those of our elderly participants. For example, we explained the mechanism of association with low BMI and dysphagia, reverse epidemiology, and the association with dementia and dysphagia using the literature. Then, we compared the results of this study as follows:

 “Our participants were older; approximately one third of the patients were aged > 80 years. The elderly population is easy to be in “frail” situation. Which is known as a state of vulnerability to poor homeostasis resolution after a stressor event [25]. Hospitalization may be a stressor to our older participants, and the patients were more likely to be in a vicious cycle of malnutrition related to dysphagia.” (Page 15, lines 9–13)

 “PEW evokes disordered catabolism, thus, leading to muscle loss [28]. Malnutrition and muscle loss affect cardiovascular disease negatively, which is a major cause of mortality in patients undergoing hemodialysis [29]. Additionally, as the immune system is susceptible to malnutrition, PEW can affect infectious complications negatively, which is also a major cause of mortality [30]. Our older “frail” hospitalized patients with low BMI were easy to be in PEW situation, and muscle loss leading to dysphagia was likely to occur. Moreover, dysphagia can cause further weight loss. In our participants, cardiovascular disease was the most common cause of death, followed by infectious diseases. Aspiration pneumonia was also one of the causes of death. This may suggest that a low BMI related to dysphasia was associated with death and life prognosis.” (Page 16, lines 1–11)

 “Although previous studies on reverse epidemiology have not discussed the nutritional status of the elderly, dysphagia in older patients undergoing hemodialysis might be considered one of the appearances of reverse epidemiology.” (Page 16, lines 14–16)

 “According to the description of medical records, most of the patients with dementia need assistance to eat. As stated in literature, difficulties in oral intake and self-feeding reported may have occurred in patients with dementia and these could lead to difficulty in eating independently.” (Page 17, lines 5–8)

We considered the “frail” characteristics of our elderly hospitalized patients as follows:

 “Our participants were older; approximately one third of the patients were aged > 80 years. The elderly population is easy to be in “frail” situation, which is known as a state of vulnerability to poor homeostasis resolution after a stressor event [25]. Hospitalization may be a stressor to our older participants, and the patients were more likely to be in a vicious cycle of malnutrition related to dysphagia.” (Page 15, lines 9–13)

“Our older “frail” hospitalized patients with low BMI were easy to be in PEW situation, and muscle loss leading to dysphagia was likely to occur. Moreover, dysphagia can cause further weight loss. In our participants, cardiovascular disease was the most common cause of death, followed by infectious diseases. Aspiration pneumonia was also one of the causes of death. This may suggest that a low BMI related to dysphagia was associated with death and life prognosis.” (Page 16, lines 6–11)

The authors affirm that a limitation of this study is “Evaluation of dysphagia using the KTBC in a larger sample at multiple centers is needed to examine generalizability of the results. The timing of the evaluation varied depending on the case. It would have been ideal to perform a KTBC assessment of all patients within a certain period post admission in our hospital.” About this statement, I have some suggestions/questions:

- The authors could discuss this important point of view in the “discussion” section;

Response: We apologize for the incoherent description. As our participants were limited to older patients undergoing hemodialysis, the results may have been influenced by the participants’ characteristics. It is difficult to apply the results of this study to the general elderly population, which was a limitation. We have discussed this issue in the revised manuscript as follows:

 “As our participants were limited to older patients undergoing hemodialysis, the results may have been influenced by the participants’ characteristics. Additionally, the association between the feeding and nutritional status was examined in a cross-sectional manner. To apply the results of this study to the general elderly population, evaluation of dysphagia using the KTBC in a larger sample at diverse facilities is needed.” (Page 19, line 16 – Page 20 line 6)

Additionally, as this was a retrospective study based on the clinical practice, it was difficult to standardize the timing of KTBC evaluation. This study revealed that the KTBC score was associated with a decrease in BMI over time. Therefore, it would be important to detect poor feeding status and prevent malnutrition in early hospitalization and intervene for improvement. Following the reviewer’s suggestion, we have discussed this issue in the Discussion section as follows:

“As this is a retrospective study based on clinical practice, it was difficult to standardize the timing of KTBC evaluation. It would be desirable to establish a protocol for early evaluation and intervention during hospitalization to prevent malnutrition.” (Page 19, lines 14–17)

- Considering this limitation, is it prudent to conclude that “the evaluation of dysphagia using the KTBC is highly recommended to prevent malnutrition.”? I don’t think so.

Response: We apologize for the incoherent description. 

As this is a retrospective study based on clinical practice, we would like to propose KTBC as one of the clinical assessment tools for monitoring the feeding status. Please note that we have added the following sentence to the revised manuscript:

 “As older patients undergoing hemodialysis are more vulnerable to dysphagia, the evaluation of dysphagia using the KTBC could be one of clinical assessment tools to prevent malnutrition.” (Page 20, lines 15–17)

Reviewer 2

We would like to thank Reviewer 2 for the time and effort taken to review our manuscript and for providing valuable comments and suggestions, which have considerably helped us improve our manuscript. We have made every effort to address the issues raised and to respond to all comments. The revisions are indicated in red font in the revised manuscripts. Please, find next a detailed, point-by point response to the reviewer’s comments. We hope that our revisions will meet the reviewer’s expectation.

TITLE

I think it would be interesting to clarify what the use of the word nutrition wants to convey. Nutritional status? Therefore, I suggest adjusting the title.

Response: As the reviewer suggested, “nutritional status” reflects the purpose of this study more than “nutrition”. Thus, we have shortened the title and adjusted it to convey our idea more clearly.

ABSTRACT

Adequate and concise. I suggest just mentioning in line 6 that they are elderly patients, and the percentage value in parentheses is equivalent to n=37 (line 10).

Response: Please note that we have mentioned that our participants were elderly patients. As the risk of dysphagia was evaluated against the median KTBC value, nearly half of the participants were classified as being at risk. We have added a percentage value in parentheses as follows:

 “We classified patients with lower than the median KTBC score (57 points) as being at risk for dysphagia; 37 patients (45.1%) were at risk for dysphagia.” (Page 2, lines 9–11)

INTRODUCTION

I suggest deleting the part of the third paragraph about the EAT-10 related information. It can be included in the discussion section if the authors wish. I think it would be the most appropriate session, precisely because it was not used in the study and contributed to making the introduction unnecessarily long.

Response: We have removed the EAT-10 related information to avoid making the Introduction section unnecessarily long. Additionally, we have moved the details of the KTBC to the Discussion section.

It is not usual to mention figures in the introduction section. The manuscript already has three tables and six figures. There is no need to exemplify the KTBC tool. If the authors wish to keep Figure 1, I suggest using the methods when citing the tool or presenting the study's radar chart in the results.

Response: Please note that we have removed Figure 1 from the Introduction section.

The fourth paragraph, it's confusing and repetitive. I recommend making the objectives clear to the reader, including that they are elderly patients.

Response: Please not that we have excluded duplicate statements and numbered two aims of this study to make our objectives clear. We have also emphasized that our participants were elderly, as per the reviewer’s suggestion. The added part is as follows:

 “Thus, the KTBC is non-invasive [11] and easy to use for medical staffs. However, there are no previous studies evaluating the feeding status of older patients undergoing hemodialysis with KTBC.” (Page 4, lines 4–6)

MATERIALS AND METHODS

In the Study design section, some information presented is duplicated, such as the data collection period until February 2022 (page 5, line 18; page 6, line 6; page 9, line 6), as well as for the nutritional indicators on pages 6 and 7. These can be presented in the Data collection section. The Study design section is large and with varied information, not only about the study design. I recommend including a Participants section to present the sample and eligibility criteria.

Response: We would like to apologize for the confusing description. We have reorganized the “Study design” subsection and added the “Participants” subsection to present the sample and eligibility criteria (Page 5, lines 7–15).

I did not understand the inclusion criterion of being over 20 years old. Is this not a sample of elderly patients?

Response: We would like to thank the reviewer for pointing this out. As our participants did not include patients aged ≤20 years, the inclusion criterion of age >20 years is not necessary. We have removed this part, as per the reviewer’s suggestion.

Authors should value the methods section and present missing information. In Data Collection, detail the sociodemographic and clinical characteristics collected. Nutritional indicators may or may not be part of a separate section, but blood tests conditions (e.g., fasting or not?) and analysis methods (e.g., albumin, colorimetric bromocresol green method?) must be described. Was there any care in measuring weight due to changes in blood volume and edema in this population?

Response: Our blood tests were not fasting, and the serum albumin level was analyzed by the bromocresol green method. We have added this information concerning blood tests in the Materials and Methods section as follows:

 “To examine the association with the KTBC score and nutritional indicators briefly, we focused on the nutritional indicators of the closest month at the time of KTBC evaluation. Blood tests were routinely conducted at the beginning of dialysis treatment (not fasting). Serum albumin was analyzed using the bromocresol green method. According to the protein energy wasting (PEW) criteria, reported by the International Society of Renal Nutrition and Metabolism, serum albumin levels < 3.8 g/dL indicate the risk of malnutrition [14]".” (Page 6, lines 7–13)

Unfortunately, we did not use a weight scale, which can estimate fluid volume and edema of the patients. Therefore, we have discussed this issue as a limitation as follows:

 “We used the BMI as a nutritional indicator related to dysphagia; however, we did not consider body fluid removal during the dialysis session. Therefore, the results did not reflect the body composition or edema status of patients. Additionally, as dysphagia is related to muscle capability, indicators reflecting muscle function should also have been considered.” (Page 20, lines 6–10)

Figure 2: There is no need to repeat the collection period in the third text box, as it was already presented in the first. 

Response: Please not that we have removed the duplicate information from the third box in Figure 1 (Figure 2 before correction).

If the KTBC was performed after three days of hospitalization, shouldn't the sample with hospitalization less than three days (n=25) from the second exclusion text box be in the first exclusion step? Before having your data collected?

Response: Please note that we have corrected our mistake following the reviewer’s suggestion.

Is it not possible to adopt a plausible cut-off point for the value of albumin in this population, as was done for the GNRI and the NRI?

Response: According to protein energy wasting (PEW) criteria, reported by the International Society of Renal Nutrition and Metabolism, serum albumin levels <3.8 g/dL indicates the risk of malnutrition. We have provided this information in the Material and Methods section, and showed the cut-off point in Figure 3a, as was done for GNRI and NRI. The added part is as follows:

 To examine the association with the KTBC score and nutritional indicators briefly, we focused on the nutritional indicators of the closest month at the time of KTBC evaluation. Blood tests were routinely conducted at the beginning of dialysis treatment (not fasting). Serum albumin was analyzed using the bromocresol green method. According to the protein energy wasting (PEW) criteria, reported by the International Society of Renal Nutrition and Metabolism, serum albumin levels < 3.8 g/dL indicate the risk of malnutrition [14].” (Page 6, lines 7–13)

In Statistical analysis, line 9 has a missing parenthesis.

Response: We would like to apologize for the mistake. We have corrected this in the revised manuscript.

RESULTS

In the Patient background section, Figure 2 is presented the methods and results. Some researchers and reviewers consider it as part of the method and others as a first result. I recommend that the authors choose which of the two sessions will mention in figure 2. In the results, there is no need to describe the figure step. It could be something more general.

Response: Thank you for the suggestion. We have chosen to present Figure 1 (Figure 2 before correction) in the Results session to avoid duplication. 

The first mention in the text about Table 1 does not present the main information in the text (page 9, line 14). This content is out of place on page 11 (between lines 8 and 11). I recommend making a single mention of Table 1 with the main findings presented in a general way at the beginning of the results session.

Response: We would like to apologize for the confusing description. Following the reviewer’s suggestion, we have moved Table 1 at the beginning of the Results session.

Present in the text comments on Figure 3.

Response: Thank you for the suggestion. Please note that we have added comments concerning Figure 2 (Figure 3 before correction).

ALL FIGURES AND TABLES: Whenever possible, use a full title for clarity for the reader.

Response: Thank you for the suggestion. Please note that we have reviewed all figures and tables and used full titles, as per the reviewer’s suggestion.

DISCUSSION

I recommend avoiding mentioning the tables and figures in this section.

Response: Following the reviewer’s suggestion, we have removed the unnecessary parts concerning the tables and figures from the Discussion section. For example, we have excluded the detailed result of the association with KTBC and dementia. Additionally, we have also excluded the detailed description concerning Supplementary Tables from the Discussion section. 

On page 17 (line 11), include the BMI unit and avoid using pejorative or stigmatizing language (e.g., “few were obese” by a few individuals with obesity).

Response: Please note that we have revised this part accordingly (Page 16, line 12).

Regarding reverse epidemiology, I think it is necessary to consider the other side of studies related to the topic. Many are inconsistent, some use BMI as a marker of adiposity, which does not measure central adiposity. At the same time, we know that the major cause of mortality in this population is attributed to cardiovascular disease.

Response: Please note that we have enhanced our discussion concerning the participant characteristics in the Discussion section as follows: 

 “Our participants were older; approximately one third of the patients were aged > 80 years. The elderly population is easy to be in “frail” situation, which is known as a state of vulnerability to poor homeostasis resolution after a stressor event [25]. Hospitalization may be a stressor to our older participants, and the patients were more likely to be in a vicious cycle of malnutrition related to dysphagia.” (Page 15, lines 9–13)

 “PEW evokes disordered catabolism, thus, leading to muscle loss [28]. Malnutrition and muscle loss affect cardiovascular disease negatively, which is a major cause of mortality in patients undergoing hemodialysis [29]. Additionally, as the immune system is susceptible to malnutrition, PEW can affect infectious complications negatively, which is also a major cause of mortality [30]. Our older “frail” hospitalized patients with low BMI were easy to be in PEW situation, and muscle loss leading to dysphagia was likely to occur. Moreover, dysphagia can cause further weight loss. In our participants, cardiovascular disease was the most common cause of death, followed by infectious diseases. Aspiration pneumonia was also one of the causes of death. This may suggest that a low BMI related to dysphasia associated with death and life prognosis.” (Page 16, lines 1–page 16, line 11)

Indeed, cardiovascular disease is the major cause of mortality, and a previous showed that malnutrition and muscle loss can be a risk factor for cardiovascular mortality in patients undergoing hemodialysis [29]. Infectious complications are also the major cause of mortality in this population [30]. Malnourished patients tend to have a weakened immune system and are more likely to develop infectious complications. 

As we mentioned in the manuscript, the mechanism of reverse epidemiology has not been elucidated fully; however, malnutrition associated with low BMI may affect major causes of mortality negatively. Especially, this negative relationship is likely to occur in “frail” hospitalized elderly patients. We have revised manuscript considering this point of view as follow:

“PEW can affect infectious complications negatively, which is also a major cause of mortality [30]. Our older “frail” hospitalized patients with low BMI were easy to be in PEW situation, and muscle loss leading to dysphagia was likely to occur. Moreover, dysphagia can cause further weight loss. In our participants, cardiovascular disease was the most common cause of death, followed by infectious diseases. Aspiration pneumonia was also one of the causes of death. This may suggest that a low BMI related to dysphasia associated with death and life prognosis. In our participants, the median BMI was 20.2 kg/m2; in few individuals with obesity, however, a higher BMI with better feeding status was associated with higher survival rates. Although previous studies on reverse epidemiology have not discussed the nutritional status of the elderly, dysphagia in older patients undergoing hemodialysis might be considered one of the appearances of reverse epidemiology. ” (Page 16, lines 5–16)

Regarding limitations, it is necessary to mention the limitation of the use of BMI and possibly the lack of an adequate parameter for more effective conclusions through the assessment of body composition and muscle function, whether strength or functional capacity of elderly individuals and its relationship with dysphagia.

Response: In the limitation part, we have emphasized the limitation of the utilization of the BMI as a parameter related to dysphagia as well as the necessity of using indicators that reflect body composition and muscle function. The added part is as follows:

 “We used the BMI as a nutritional indicator related to dysphagia; however, we did not consider body fluid removal during the dialysis session. Therefore, the results did not reflect the body composition or edema status of patients. Additionally, as dysphagia is related to muscle capability, indicators reflecting muscle function should also have been considered.” (Page 20, lines 6–10)

---

## [Decision Letter · Decision Letter 1]

2 Dec 2022

Importance of feeding status evaluation in older patients undergoing hemodialysis

PONE-D-22-12985R1

Dear Dr. Kitamura,

We’re pleased to inform you that your manuscript has been judged scientifically suitable for publication and will be formally accepted for publication once it meets all outstanding technical requirements.

Kind regards,

Mabel Aoun, MD, MPH

Academic Editor

PLOS ONE

Additional Editor Comments (optional):

Reviewers' comments:

Reviewer's Responses to Questions

**Comments to the Author**

1. If the authors have adequately addressed your comments raised in a previous round of review and you feel that this manuscript is now acceptable for publication, you may indicate that here to bypass the “Comments to the Author” section, enter your conflict of interest statement in the “Confidential to Editor” section, and submit your "Accept" recommendation.

Reviewer #1: (No Response)

Reviewer #2: All comments have been addressed

2. Is the manuscript technically sound, and do the data support the conclusions?

Reviewer #1: (No Response)

Reviewer #2: Yes

3. Has the statistical analysis been performed appropriately and rigorously? 

Reviewer #1: (No Response)

Reviewer #2: Yes

4. Have the authors made all data underlying the findings in their manuscript fully available?

Reviewer #1: (No Response)

Reviewer #2: Yes

5. Is the manuscript presented in an intelligible fashion and written in standard English?

Reviewer #1: (No Response)

Reviewer #2: Yes

6. Review Comments to the Author

Reviewer #1: (No Response)

Reviewer #2: I would like to congratulate the authors of the manuscript. As well as for the effort to solve my doubts, suggestions, and questions raised.

The only information that I was not able to identify in the revised version was the age adopted as a criterion for inclusion in the sample of elderly patients on hemodialysis. I think it's important for the reader.

7. PLOS authors have the option to publish the peer review history of their article (what does this mean?). If published, this will include your full peer review and any attached files.

Reviewer #1: No

Reviewer #2: No

---

## [Editor Report · Acceptance letter]

22 Dec 2022

PONE-D-22-12985R1 

Importance of feeding status evaluation in older patients undergoing hemodialysis 

Dear Dr. Kitamura:

I'm pleased to inform you that your manuscript has been deemed suitable for publication in PLOS ONE. Congratulations! Your manuscript is now with our production department. 

Kind regards, 

on behalf of

Dr. Mabel Aoun 

Academic Editor

PLOS ONE